# Sleep Disorders and Psychological Profile in Oral Cancer Survivors: A Case-Control Clinical Study

**DOI:** 10.3390/cancers13081855

**Published:** 2021-04-13

**Authors:** Roberta Gasparro, Elena Calabria, Noemi Coppola, Gaetano Marenzi, Gilberto Sammartino, Massimo Aria, Michele Davide Mignogna, Daniela Adamo

**Affiliations:** 1Department of Neurosciences, Reproductive Science and Dentistry, Federico II University of Naples, Via Pansini 5, 80138 Naples, Italy; roberta.gasparro@unina.it (R.G.); noemi.coppola@unina.it (N.C.); gaetano.marenzi@unina.it (G.M.); gilberto.sammartino@unina.it (G.S.); mignogna@unina.it (M.D.M.); danielaadamo.it@gmail.com (D.A.); 2Department of Economics and Statistics, Federico II University of Naples, Via Cinthia, Monte Sant’Angelo, 80126 Naples, Italy; aria@unina.it

**Keywords:** oral cancer, sleep disturbance, depression, anxiety, insomnia, oral cancer survivors, psychiatric profile

## Abstract

**Simple Summary:**

Sleep disorders have been increasingly investigated in several medical illnesses as their presence may affect patients’ quality of life. However, the research examining sleep disorders in oral cancer is relatively weak. Indeed, the majority of the available studies present a cross-sectional or retrospective designs. Moreover, very few of them have evaluated quality of sleep in oral cancer survivors (OC survivors). We aimed to carry out a case-control study with the purpose to investigate sleep disorders and mood impairment in 50 OC survivors. Our research has shown that quality of sleep is significantly affected in OC survivors compared to a healthy population and that OC survivors suffers from higher levels of anxiety and depression. Our results may suggest that an appropriate assessment of quality of sleep and psychological profile should be performed in OC survivors as a prompt treatment for both sleep and mood disorders is crucial for the overall improvement of patients’ quality of life.

**Abstract:**

Quality of sleep (QoS) and mood may impair oral cancer survivors’ wellbeing, however few evidences are currently available. Therefore, we aimed to assess the prevalence of sleep disorders, anxiety and depression among five-year oral cancer survivors (OC survivors). 50 OC survivors were compared with 50 healthy subjects matched for age and sex. The Pittsburgh Sleep Quality Index (PSQI), the Epworth Sleepiness Scale (ESS), the Hamilton Rating Scales for Depression and Anxiety (HAM-D, HAM-A), the Numeric Rating Scale (NRS), the Total Pain Rating Index (T-PRI) were administered. The global score of the PSQI, ESS, HAM-A, HAM-D, NRS, T-PRI, was statistically higher in the OC survivors than the controls (*p*-value: <0.001). QoS of OC survivors was significantly impaired, especially with regard to some PSQI sub-items as the subjective sleep quality, sleep latency and daytime dysfunction (*p*-value: 0.001, 0.029, 0.004). Moreover, poor QoS was negatively correlated with years of education (*p*-value: 0.042 *) and positively correlated with alcohol consumption (*p*-value: 0.049 *) and with the use of systemic medications (*p*-value: 0.044 *). Sleep disorders and mood disorders are common comorbidities in OC survivors; therefore, early assessment and management before, during and after treatment should be performed in order to improve the quality of life of OC survivors.

## 1. Introduction

Oral cancer is a life-threatening disease and a burden for health care systems worldwide. According to Global Cancer Statistics, GLOBOCAN, there were 354,864 new cases of oral cavity cancer causing 177,384 deaths during 2018 [1]. Despite the improvement in diagnosis and treatment by health care providers with a subsequent decrease in mortality, the quality of life of oral cancer survivors (OC survivors) remains poor on account of the impact of this disease on mental and emotional well-being. Indeed, oral cancer patients often suffer from emotional distress, fatigue, sleep disturbance, anxiety and depression that can arise during treatment and persist long-term, aggravating the burden of the disease [2,3].

Recently, a growing interest has been focused on the evaluation of sleep disorders in relation to several medical illnesses as their presence may worsen the underlying disease and increase the rate of mortality [4]. Furthermore, sleep disorders are considered to be an extremely sensitive marker for psychiatric comorbidities which may also precede mood disorders, especially depression or anxiety, and its early detection and treatment is crucial to improve the prognosis and quality of life of patients.

Insomnia is the most frequent sleep disorder; generally, patients report a difficulty in falling asleep, and often experience restless sleep and excessive daytime sleepiness (hypersomnolence) [5].

The overall incidence of insomnia in cancer patients has been found to be three times higher than that reported in the general population and ranges from 30.0% to 93.1%, depending on the type of cancer [6,7].

This high incidence is probably related to a post-diagnosis experience marked by a series of stressors that can act as a trigger for insomnia and, if they persist, may contribute to a chronic development causing long-lasting sleep disturbance even after the cancer treatment ends.

In a recent systematic review, the prevalence of insomnia in oral cancer patients was 29.0% before, 45% during and 40% after the treatment while hypersomnolence was reported by 16% and 32% of patients before and after the treatment, respectively [8].

The persistence of sleep disorders such as insomnia and hypersomnolence may negatively affect the quality of life of OC survivors and has a powerful influence on the increased risk of infectious disease, and on the occurrence and progression of several major medical illnesses including cardiovascular diseases and mood disorders [9]. Sleep disorders activate biological mechanisms, such as inflammation which are increasingly thought to contribute to depression, and potentially increase the risk of cancer morbidity and related mortality [10]. Indeed, sleep duration has been closely related to a poor overall survival and cancer-specific death over a ten-year follow-up period [11].

In contrast to the substantial literature on depression, research examining sleep disorders in oral cancer is relatively weak, with the majority of studies using a cross-sectional or retrospective analysis. In addition, most of the studies have evaluated the prevalence of sleep disorders before the start or during the treatment while very few studies have included OC survivors in follow-up. Moreover, the role of predictors in sleep disorders remains unclear.

Therefore, we have designed a case-control study to better evaluate the difference in the prevalence of sleep disorders between OC survivors and healthy subjects. The purposes of this study were: (1) to investigate the prevalence of sleep disorders (insomnia and daytime sleepiness), pain, anxiety and depression among OC survivor patients, (2) and to evaluate the potential predictors of sleep disorders such as socio-demographic data, habits, body mass index (BMI), pain, anxiety, depression, medical comorbidities and drug intake and the staging and grading of the oral cancer.

## 2. Material and Methods

### 2.1. Study Design and Participants

A case-control study was carried out at the Oral Medicine Department of Federico II University of Naples in accordance with the ethical principles of the World Medical Association Declaration of Helsinki. The study was approved by the Research Ethics Committee (protocol number 188: 2014). The methods adopted conformed with the Strengthening the Reporting of Observational Studies in Epidemiology (STROBE) guidelines for observational studies (Appendix A) [12].

The recruitment of OC survivors and healthy subjects was conducted between January and September 2018 and was based upon convenience sampling. All potentially eligible individuals were invited to participate in the present study and provided their written informed consent.

The case and the control groups were matched by age and gender. Specifically, first we recruited the patients and then calculated the gender distribution and the average age; secondly, we recruited the controls to obtain a matched sample.

Participants of either gender and aged 18 or older were included. The inclusion criteria for the OC survivors’ group were: (i) clinical and histopathological findings of oral squamous cell carcinoma (OSCC) or tobacco-related verrucous cell carcinoma (VCC) (ii) patients with a follow-up of at least five years after the diagnosis of OSCC or VCC and being free from malignancy for at least one year, (iii) all stages based on the American Joint Committee on Cancer Staging Manual 8th edition and (iv) patients managed by surgery, radiotherapy and/or chemotherapy.

On the contrary, the exclusion criteria for the case group were: (i) patients affected by human papillomavirus (HPV)-related OSCC, (ii) patients affected by another type of tumor localized at the head and neck region, (iii) patients who had concomitant tumors in another organ, and (iv) patients who had experienced severe and irreversible side effects from OSCC treatment such as fibrosis, a mouth opening restriction of less than 30 mm, trismus, hyposalivation or osteoradionecrosis of the jaw.

The inclusion criteria for the control group were: (i) patients treated at the University Dental Clinic only for routine dental care during the study period; and (ii) the absence of any oral mucosal lesions or any previous history of OSCC/VCC.

For both groups the exclusion criteria were (i) breastfeeding or pregnant participants, (ii) patients affected by autoimmune disease or another debilitating condition or unstable disease (such as osteonecrosis of the jaw or dementia), (iii) participants with a medical history of a psychiatric disorder as defined by the DSM-5 or regularly treated with a psychotropic drug, (iv) drug-addicted or alcoholic participants and (v) individuals unable or not willing to give their consent or to understand and complete the questionnaires.

### 2.2. Procedure

A comprehensive intra- and extra-oral examination was carried out by two oral medicine experts (RG and AD). Upon admission, demographic data such as gender, age, educational level (in years), marital status, employment status, risk factors (smoking and alcohol consumption) body mass index (BMI), comorbidities and associated drug use were recorded for both groups.

Details of clinical oral cancer related characteristics were also noted for the case-group, such as the clinical stage and grading at the time of diagnosis, the location of the tumor, any clinical nodal involvement, any metastasis, the type of treatment, and any need for further treatment during the 5-year follow-up. The performance status was assessed using the Eastern Cooperative Oncology Group (ECOG) scale in OC survivors whose scores range from 0 (fully active) to 5 (death), with higher values indicating a poorer performance status [13].

A predefined set of questionnaires was given to the participants of both groups in order to assess their quality of sleep (QoS), their psychological status (level of anxiety and depression) and the intensity and quality of any pain. The questionnaires comprised:-the Pittsburgh Sleep Quality Index (PSQI) [14] for the evaluation of insomnia;-the Epworth Sleepiness Scale (ESS) [15] for the assessment of hypersomnolence;-the Hamilton Rating Scale for Depression (HAM-D) [16] and the Hamilton rating scale for Anxiety (HAM-A) [17] to evaluate depression and anxiety, respectively;-the Numeric Rating Scale (NRS) [18] and the short form of the McGill Pain Questionnaire (SF-MPQ) [19] for the evaluation of the intensity and quality of any pain. All the questionnaires were administered in their Italian version and were reviewed for completeness before collection.

### 2.3. Outcome Measures

#### 2.3.1. Measures of the Quality of Sleep

The Pittsburgh Sleep Quality Index (PSQI) is a standardized questionnaire used for the assessment of the QoS and the incidence of sleep disturbances. This tool consists of 19 items which generate 7 ‘component’ scores: subjective sleep quality, sleep latency, sleep duration, habitual sleep efficiency, sleep disturbances, use of sleep medication and daytime dysfunction. The scores for each item range from 0 to 3, with higher scores indicating a poorer QoS. The items are combined to yield the seven components, each component having a score ranging from 0 to 3, and the sum of the scores for these seven components yields a global score ranging from 0 to 21. Global scores above five distinguish poor sleepers from good sleepers with a high sensitivity (90–99%) and specificity (84–87%) [14].

The Epworth Sleepiness Scale (ESS) is used to measure an individual’s general level of daytime sleepiness. The tool consists of 8 items assessing the propensity for sleep in eight common situations. Subjects rate their likelihood of dozing in each situation on a scale of 0 (would never doze) to 3 (would have a high chance of dozing). The ESS score is the sum of the eight items, ranging from 0 to 24, with a cut-off value of >10 indicating excessive daytime sleepiness [15].

#### 2.3.2. Measures of Psychological Factors

The Hamilton Rating Scale for Anxiety (HAM-A) is a measure of symptoms of anxiety and it consists of 14 items. Scores can range from 0 to 56, with scores from 7 to 17 indicating mild symptoms, between 18 and 24 indicating mild-to-moderate severity, and >25 indicating moderate-to-severe anxiety [16].

The Hamilton Rating Scale for Depression (HAM-D) is a measure of symptoms of depression that is comprised of 21 items pertaining to the affective field. Scores can range from 0 to 54. Scores between 7 and 17 indicate mild depression, between 18 and 24 moderate depression, and over 24 severe depression [17].

#### 2.3.3. Measures of Pain

The Numeric Rating Scale (NRS-11) is a well-validated instrument for the evaluation of pain intensity. whose scale ranges from 0 to 10 (0 = no oral symptoms and 10 = the worst imaginable discomfort). Respondents are asked to report pain intensity in the last 24 h [18].

The Total Pain Rating Index (T-PRI) from the short form of the McGill Pain Questionnaire (SF-MPQ) is a measure of the quality of pain and it is a multidimensional pain questionnaire which measures the sensory, affective and evaluative aspects of the perceived pain. It comprises 15 items from the original MPQ, each scored from 0 (none) to 3 (severe). The T-PRI score is obtained by summing the item scores (range 0–45). There are no established critical cut-off points for the interpretation of the scores and, as for the MPQ, a higher score indicates worse pain [19].

### 2.4. Statistical Analysis

Descriptive statistics, including means, standard deviations, medians and the inter-quartile range (IQR) were used to analyse all the socio-demographic and clinical characteristics of the two groups. For the qualitative variables, the significance was calculated by the Exact Chi Square Test. For the demographic numerical variables the significance difference between means was calculated by the parametric two-samples *t*-test procedure. The significance difference between the recorded medians of the PSQI, ESS, HAM-D, HAM-A, NRS and T-PRI, was measured by the Mann-Whitney Test.

The addition of the clinical characteristics predictors of a poor QoS in OC survivors, hierarchical multiple regression analyses were performed and unadjusted coefficient estimations were obtained for each predictor. A total of six models was computed. The coefficient estimated for binary variables, such as smoking and alcohol consumption, measures the effect of the Yes response on the outcome estimation. For each model, we reported the adjusted R2 which measures the overall goodness of fit adjusted for the number of variables included into the model. The demographic model (model 1) was performed to test the contribution of the demographic variables to a poor QoS. Next, the clinical model (model 2), the psychological model (model 3), the daytime sleepiness model (model 4) and the pain model (model 5) were each performed after controlling for demographic variables to test the contribution of the clinical variables of the OSCC, anxiety and depression (HAM-A; HAM-D), daytime sleepiness (ESS), intensity and quality of pain (NRS, T-PRI) to a poor QoS. Finally, a standard regression analysis (model 6) was computed by entering all the variables simultaneously into the model in order to determine the relative contributions of all the variables to a poor QoS. In all the steps, standard errors of the model coefficients, which measure the statistical precision of the inference estimation of the model parameters, were provided. The IBM SPSS version 22.0 was used to conduct all the statistical analyses in this study, and *p*-value < 0.05 (two-tails) was considered as statistically significant.

## 3. Results

The demographic characteristics, BMI and habits of the case and control groups are summarized in Table 1. A total of 100 participants were included in this study, 50 OC survivors and 50 healthy participants and no missing data were recorded.

Of these participants, 54% (*n* = 26) and 46% (*n* = 24) were male and female for each group, respectively, with a mean age of 59.5 ± 10.1 years for the cases and 65.1 ± 14.4 years for the controls (*p*-value: 0.051). No statistically significant difference was found in terms of marital status, years of education, BMI or alcohol consumption (*p*-values: 0.115, 0.054, 0.068, 0.619, respectively). However, the number of healthy participants in full-time employment and with a current smoking habit was significantly higher (*p*-value: <0.001 ** and 0.005 *** respectively) in comparison to the case group.

Table 2 shows the prevalence of systemic diseases and drug intake in the study sample. The OC survivors presented with a statistically higher number of systemic comorbidities in comparison to the control group (*p*-value: 0.012 *), especially with respect to hypertension, hypercholesterolemia, prostatic hypertrophy and gastrointestinal diseases (*p*-values: <0.001 **, 0.001 **, 0.16 * and <0.001 *** respectively). Consequently, the number of OC survivors taking medications, such as angiotensin II receptor antagonists, beta blockers, proton pump inhibitors and statin agents was significantly higher compared to the controls (*p*-value: <0.001 **).

Table 3 summarizes the clinical characteristics of the OC survivors. The majority of the patients were diagnosed with stages 0–1 (52%) while 48% were diagnosed with stages 3–4 and with differentiated OSCC (G1-2 88% of the patients). Most of the tumors were localized at the tongue (52%) and alveolar ridges (22%), while 16% and 10% at the buccal mucosa and hard/soft palate, respectively. All the patients with OSCC were managed with surgical treatments ranging from local conservative tumor excision (66.0%) to more invasive surgical treatments. such as hemiglossectomy (20%), maxillary osteotomy (8.0%), hemimandibulectomy (6%) and cervical neck dissection (42%). Only a few patients received, in addition, radiotherapy (16%) or chemotherapy (2%). Tracheostomy was not performed in respect of any OC survivors. Overall, the OSCC patients were further treated with incisional or excisional biopsies over the five-year follow-up period (a mean of 4.8 +/− 2.9) due to local relapses, especially in respect of the 29 (58%) OC survivors with associated potentially malignant disorders such as lichenoid lesions 8 (16%), leukoplakia 7 (14%) erythroleukoplakia 14 (28%).

At the time of the assessment, 66% of the OSCC patients presented with an ECOG performance status of 0 (“fully active”) and 34% with an ECOG performance status of 1 (“restricted in physically strenuous activity”).

Among the OC survivors, 52% were poor sleepers (PSQI > 5), whereas only 12% of the controls reported a poor QoS. Moreover, mild to severe anxiety was reported in 84% of the OC survivors (48% mild, 12% moderate and 24% severe anxiety) along with mild to severe depression in 74% of cases (40% mild, 16% moderate and 18% severe depression). On the contrary, only 20% and 18% of the healthy participants showed mild anxiety and depression symptoms, respectively, and no cases of moderate to severe anxiety or depression were recorded in the control group.

Table 4 shows the differences in all the psychological factors between the case and control group. A Cronbach alpha value of 0.76 and 0.91 was indicative of a good reliability of the PSQI scale in both groups. The OC survivors presented a mean of hours of sleep of 6.94 ± 1.024, while the controls slept a mean of 7.16 ± 0.681 h. A statistically significant difference was found between the medians of all the psychological variables assessed in terms of QoS, anxiety and depression and intensity and quality of pain. The OC survivors showed statistically significant higher scores in the global PSQI (*p*-value: 0.017 *), especially for the items “subjective sleep quality”, “sleep latency” and daytime dysfunction” (*p*-values: <0.001 **, 0.029 * and 0.004 ** respectively), and in the total ESS score (*p*-value: 0.001 **) in comparison with the controls. Furthermore, statistically significant higher levels of anxiety and depression, as reflected by the total scores of the HAM-A and HAM-D, were also recorded among the OC survivors (*p*-value: <0.001 **), together with higher levels of oral discomfort and pain according to the NRS and T-PRI total scores (*p*-value: 0.001). Taken together, these findings suggest that QoS and psychological status may be severely impaired in OC survivors.

Furthermore, in the case group, a statistically significant positive correlation was found between the global PSQI score and the HAM-A, HAM-D and T-PRI scores (*p*-values: <0.001 **, <0.001 ** and 0.019 * respectively) but not with the ESS and the NRS. Specifically, the majority of the PSQI sub-items (except for “use of sleep medication” and “sleep latency”) were positively correlated with the HAM-A and HAM-D (except for “use of sleep medication”), whereas the T-PRI was correlated only with “sleep disturbances and daytime dysfunction” which also correlated, as expected, with the ESS. Overall, patients with a poorer QoS presented with higher levels of anxiety and depression and a worse quality of pain but not with increasing daytime sleepiness or pain intensity (Table 5).

The hierarchical multiple regression analyses predicting QoS are shown in Table 6. The first model (the demographic model), testing the contribution of demographic variables and risk factors (alcohol and smoking) to QoS, showed that the PSQI was negatively correlated with years of education (*p*-value: 0.042 *) and resulted in a strongly significant increase in the coefficient of determination (R2) (∆R2 = 31.7%, *p*-value: 0.009). The addition of the clinical characteristics showed that the PSQI was positively correlated with alcohol consumption (*p*-value: 0.018 *) and with the use of systemic medications (*p*-value: 0.045 *). When entering all the variables simultaneously in the second model, we found an increase in the R2 value with a ∆R2 of 6.2%, possibly due to both the parameters, namely alcohol consumption and medications, although it was not statistically significant (*p*-value: 0.222). The third model (the psychological model), testing the contribution of anxiety and depression to QoS, showed that the PSQI was positively correlated with the HAM-A and HAM-D (*p*-value: 0.001 **) and resulted in a strongly significant increase in the R2 (∆R2 = 20.4%, *p*-value: <0.001 **). The daytime sleepiness and pain models (models 4 and 5) did not result in a significant increase in the R2 value (∆R2 = −2.1%, 0.0%; *p*-value: 0.749 and 0.377 respectively). The final full model (model 6, the standard multiple regression analysis) in which all of the variables were entered simultaneously (including demographic variable, risk factors, clinical characteristics, medications, anxiety, depression, daytime sleepiness, pain) resulted in a moderate increase in the R2 value (∆R2 = 12.6 %; *p*-value: 0.043 *) and could explain the 44.3% of variance of poor QoS. In this last model, depression has shown a strong correlation to sleep disorders (*p*-value: 0.001 **) contributing significantly to a poor QoS.

## 4. Discussion

The aim of this study has been to investigate the prevalence of sleep disorders (insomnia and hypersomnolence), anxiety and depression in OC survivors with a 5-year follow-up and to analyze potential predictors in the development of sleep disorders. The detection and treatment of factors which could influence the well-being of OC survivors are becoming increasingly important for healthcare systems in order to improve the follow-up care of these patients.

Among this population, insomnia, poor QoS, short sleep duration, excessive daytime sleepiness and sleep-related breathing are commonly reported and tend to become often chronic and pervasive in patients during and after treatment for OSCC [3].

In a recent systematic review, the prevalence of self-reported insomnia (defined with a PSQI cut-off of 5) in patients with head and neck cancer was 29% before treatment, 45% during treatment and 40% after treatment, while the prevalence rate of hypersomnolence (ESS cut-off > 10) was 16% before and 32% after treatment [8].

In this study, a higher prevalence of insomnia among the OC survivors within the 5-year follow-up was found, in comparison with the study of Santoso et al. [8] as 52% of the patients were poor sleepers (median PSQI score 6), while hypersomnolence was found in 24 % of OC survivors, in line with previous research [20,21].

With regard to the PSQI components a higher percentage of OC survivors reported an impaired subjective sleep quality, sleep latency, and daytime dysfunction.

Pain, fatigue, medical treatment, psychological profile (anxiety and depression) and comorbidities [22] may cause poor sleep in cancer patients. In this study, the full model of the multiple regression analysis, where all the variables were entered simultaneously, could explain only 44.3% of the variance of the PSQI in OC survivors, suggesting that the occurrence of insomnia could be independent of the cancer characteristics, staging of the malignancy, type of treatment (surgery, or radiotherapy), pain and presence of potentially malignant disorders. Instead, poor sleep was negatively correlated with years of education and positively correlated with mood disorders (anxiety and depression), the use of systemic medications and the consumption of alcohol. Therefore, a lower education level, the use of systemic drugs, the consumption of alcohol and the presence of anxiety and depression were predictors for poor sleep in OC survivors.

In a previous study, a lower education level, the presence of systemic comorbidities and the use of systemic drugs, adversely affected quality of life outcomes in survivors of cancer [23]. Moreover, there is evidence that sleep disorders may be associated with cardiovascular diseases and cardiovascular risk factors, such as hypertension and elevated resting heart rate in the general population [24], and that cardiovascular medications such as beta adrenergic blocking agents, ACE inhibitors, calcium channel antagonists may negatively affect sleep quality in individuals with other comorbidities, especially those with sleep disorders breathing [25].

Our results are in line with these studies, suggesting that the use of medications for systemic comorbidities could have a detrimental effect on the life of patients that over time could also influence QoS. However, medications with alcohol consumption contributed to sleep disorders on the account of 6.2% of the variance of poor QoS based on the second model of the regression analysis which suggests that medications may not have a pivotal role in explaining the higher prevalence of sleep disorders in this group of OC survivors, possibly for the absence of sleep disorder breathing and obstructive sleep apnea in our sample.

In addition, the low intensity of pain (NRS: 2) reported by OC survivors is considered as a predictor of poor sleep, as suggested by the regression analysis. Although xerostomia was not detected in our sample of patients probably because radiotherapy was prescribed in only 16% (8) of patients, Shuman et al [26] similarly reported that pain in the mouth and xerostomia (dry mouth) were strong predictors of poor sleep.

Regarding habits, alcohol abuse and tobacco smoking might play a role in the development of sleep disorders. Indeed, heavy alcohol users often experience insomnia even after they stop their alcohol consumption, while smokers suffer more frequently from poor sleep, compared with non-smokers [27,28]. In this study, at the time of evaluation, only 16% (8) were current smokers, as the majority had stopped their smoking habit after their OSCC diagnosis. Conversely, 42% (21) continued to consume alcohol (<14 units per week), although no one was a heavy drinker. Therefore, the positive correlation between poor sleep and alcohol consumption could be related to a previous higher alcohol consumption.

While in a recent study insomnia and hypersomnolence were found to be associated with chemotherapy and radiotherapy, [23] in the present study we could not find this correlation, presumably because the majority of the patients were in stage 0/1 (52%, 26 individuals) and only 2% (1) and 16% (8) of patients, respectively, had received these protocols. A recent review article suggested that surgery may have a positive effect on sleep quality; indeed, patients with oral cancer treated with surgery were less prone to develop insomnia, probably because they considered the operation as a resolution of the disease. The authors found a prevalence of insomnia of 31.9% in oral cancer patients who had undergone surgery and of 44.9% in those who were not receiving surgery, especially females. An explanation of these results could be that women are more vulnerable to the stress related to a cancer diagnosis and subsequently to mood disorders on account of their hormonal status [29]. In the current study we did not find any differences between male and female OC survivors, all the patients having been treated with surgical procedures.

Previous studies have suggested that obesity (BMI > 30) is considered a significant predictor of sleep disorders [30]. In our study, only 16% (8) of OC survivors were overweight, however, based on the result of the regression analyses, BMI may not have contributed to sleep disorders, similarly to the findings from the study of Bardewell et al [31].

Regarding the psychological profile, the current literature has reported a prevalence of anxiety and depression, ranging from 19 to 50%, in cancer survivors, suggesting that the burden of cancer diagnosis and its treatment could have a strong impact on the psychological profile, persisting over time despite a successful operation and subsequently decreasing the quality of life of the affected patients. Moreover, Espie et al. reported that from 22% to 32% of OC survivors were anxious or depressed even ten years after the diagnosis and treatment [32]. Factors identified as contributing to an increased risk of psychological distress among oral cancer patients include persistent pain, age (generally, younger patients more seriously affected than older patients), gender (females more seriously affected than males), stage of cancer, type of treatment, and fear of cancer recurrence. Moreover, anxiety and oral dysfunction, including trismus, xerostomia, sticky saliva and problems with eating and social contacts, are also considered a barrier to any return to work after treatment among head and neck cancer survivors [33]. As a consequence, a lack of full-time employment can exacerbate the depressive symptoms.

In this study, a higher prevalence of mood disorders has been found in comparison with the current literature; indeed, anxiety and depression were identified in 84% (42%) and 74% (37) of OC survivors, respectively. In addition, in the final full model, depression was found to be the most contributive factor to poor QoS. The higher level of depression may be related to the stress associated with a fear of cancer recurrence, since almost 40% [3] of patients presented a local cancer recurrence and, therefore, underwent a subsequent operation during the five years of follow-up.

Mood disorders and poor sleep were closely interconnected, as shown by the correlation analysis. In addition, anxiety and depression were predictors of poor sleep, as confirmed by the regression analysis. No differences between male and female patients were detected, and neither the stage and treatment nor the number of operations for cancer recurrence affected the incidence of sleep disorders. In line with previous studies, an impaired mood and sleep affected the functional recovery of patients and their return to work because, despite their age, the majority of OC survivors (48%) had retired.

The results of this study suggest that the high prevalence of insomnia may be related not only to psychiatric symptoms or to a fear of cancer recurrence but could also be considered in some cases an independent variable (as shown by the regression analysis) which needs to be addressed regardless of all the other factors. It is possible to consider that cancer itself can lead to the development of sleep disorders through inflammation. Inflammation has emerged as a crucial pathway which may be especially relevant with respect to cancer survivors. The sleep-wake cycle has emerged as a homeostatic regulator of inflammatory biology in which sleep loss induces an activation of nuclear factor KB (NF-kb) [34] and circulating levels of IL-6 [35], which coordinate the production of inflammatory mediators and systemic inflammation. In turn, pro-inflammatory cytokines are thought to contribute in part to the onset of depressive symptoms, which can amplify sleep disorders [36,37]. Moreover, chronic inflammation may predispose to a second primary recurrence [38].

Adequate sleep is a biological requirement for healthy physical, cognitive and psychological functioning so the management of sleep disturbance should be targeted by clinicians with appropriate interventions. In particular, the prominent role of cognitive behavior therapy has been studied [39]. * Additionally, the administration of melatonin in relation to the management of the sleep-wake cycle and mood disturbance as well as with respect to the quality of life of cancer patients has been proposed [40].

The findings of the current study should be understood in the light of some limitations. First, the sample is small and all the patients were recruited at a single hospital, thus preventing the possibility of any geographical generalizability and slightly affecting the power of the regression analyses. Secondly, the exclusion of patients who had developed severe and permanent side effects due to the radiotherapy, may have produced a potential underestimation of the prevalence of sleep disorders in OC survivors. Moreover, the study design does not allow the drawing of any conclusive inferences about the temporality and causality of the relationships between the variables explored. Finally, only subjective sleep quality was investigated in this study, with objective sleep quality not being considered, and therefore additional measurement systems should be incorporated to verify our findings.

## 5. Conclusions

Sleep disorders (including insomnia and hypersomnolence) continue to be prevalent both during and after treatment for OSCC. A lower level of education, the use of systemic drugs, the consumption of alcohol and the presence of anxiety and especially depression are predictors of poor sleep in OC survivors.

The treatment of oral cancer must clearly remain the major goal, but the treatment of any psychological comorbidities is also important in order to improve the quality of life in these patients. Therefore, healthcare professionals should be encouraged to include sleep disorders assessment at the time of diagnosis, during treatment and in follow-up consultations. Further clinical and prospective studies should be conducted not only to evaluate the real prevalence of sleep disorders but also to plan an adequate treatment over time with respect to all OC survivors.

## Figures and Tables

**Table 1 cancers-13-01855-t001:** Socio-demographic profile, body mass index, disease onset, and risk factors in the 50 OC survivors and 50 controls.

	OC Survivors	Controls	
	Mean ± SD	Mean ± SD	*p*-Value
Age	59.5 ± 10.1	65.1 ± 14.4	0.051
Years of education	8.5 ± 3.0	10.3 ± 5.0	0.054
	N° (%)	N° (%)	
Gender M:F	26:24 (52%, 48%)	26:24 (52%, 48%)	1.00
Marital status (married)	33 (66%)	40 (80%)	0.115
Full-time employment			<0.001 **
Employed	14 (28.0%)	36 (72.0%)	
Not employed	12 (24.0%)	8 (16.0%)	
Retired	24 (48.0%)	3 (12.0%)	
BMI			0.068
<16.5	1 (2.0%)	0 (0.0%)	
16.5–18.4	1 (2.0%)	0 (0.0%)	
18.5–24.9	19 (38.0%)	29 (58.0%)	
25.0–29.9	21 (42.0%)	21 (42.0%)	
30.0–34.9	5 (10.0%)	0 (0.0%)	
35.0–39.9	3 (6.0%)	0 (0.0%)	
≥40.0	0 (0.0%)	0 (0.0%)	
Mean ± SD	26.1 ± 4.6	27.4 ± 1.8	
Smoking	9 (18.0%)	23 (46%)	0.005 **
Alcohol consumption	21 (42.0%)	18 (36.0%)	0.619

The significance difference between means was measured by the *t*-student test. The significance difference between the percentages was measured by the Pearson Chi Square test. * Significant 0.01 < *p* ≤ 0.05, ** Significant *p* ≤ 0.01. Legend: BMI = body mass index; OSCC = oral squamous cell carcinoma.

**Table 2 cancers-13-01855-t002:** Frequency of systemic diseases and drug consumption in the 50 OSCC patients and 50 controls.

	OC Survivors	Controls	*p*-Value
	N° (%)	N° (%)	
SYSTEMIC DISEASES	37 (74.0)	24 (48.0)	0.012 *
Hypothyroidism	5 (10.0)	14 (7.0)	0.244
Hyperthyroidism	3 (6.0)	8 (16.0)	0.084
Hypertension	26 (52.0)	9 (18.0)	0.001 **
Hypercholesterolemia	22 (44.0)	3 (6.0)	<0.001 **
Previous Heart Attack	2 (4.0)	2 (4.0)	0.457
Arrhythmia	7 (14.0)	2 (4.0)	0.074
HCV +	2 (4.0)	0 (0.0)	0.437
Other hepatitis	0 (0.0)	0 (0.0)	1.000
Type 2 diabetes	3 (6.0)	0 (0.0)	0.189
Type 1 diabetes	3 (6.0)	0 (0.0)	0.189
Other cancer	3 (6.0)	0 (0.0)	0.189
Prostatic hypertrophy	5 (10.0)	0 (0.0)	0.016 *
Gastro-intestinal disease	9 (8.0)	0 (0.0)	<0.001 **
Respiratory illness	2 (4.0)	0 (0.0)	0.189
Other	2 (4.0)	0 (0.0)	0.189
DRUG CONSUMPTION			
ACE inhibitors	8 (16.0)	0 (0.0)	<0.001 **
Antiplatelets	12 (24.0)	5 (10.0)	0.010 **
Anticoagulants	3 (6.0)	0 (0.0)	0.189
Beta adrenergic blocking agents	14 (28.0)	3 (6.0)	<0.001 **
Biphosphonates	2 (4.0)	0 (0.0)	0.438
CCB (calcium channel antagonists)	5 (10.0)	0 (0.0)	0.034 *
Diuretics	9 (18.0)	4 (8.0)	0.026 *
Proton pump inhibitors	14 (28.0)	0 (0.0)	<0.01 **
Insulin	3 (6.0)	0 (0.0)	0.189
Hypoglycemic agents	3 (6.0)	0 (0.0)	0.189
Levothyroxine	4 (8.0)	12 (24.0)	0.017 *
ARB (angiotensin II receptor antagonists)	14 (28.0)	4 (8.0)	0.004 **
Statins	18 (36.0)	3 (6.0)	<0.001 **
Other drugs	0 (0.0)	1 (2.0)	0.478

The significance difference between percentages was measured by the Pearson Chi Square test. * Significant 0.01 < *p* ≤ 0.05, ** Significant *p* ≤ 0.01.

**Table 3 cancers-13-01855-t003:** Medical characteristics of the OC survivors.

OC Survivors	N° (%)
TUMOR TYPE	
Squamous cell carcinoma	47 (94.0)
Verrucous cell carcinoma	3 (6.0)
TUMOR LOCALIZATION	
Tongue and mouth floor	26 52.0)
Alveolar ridge and gingiva	11 (22.0)
Buccal mucosa	8 (16.0)
Soft and hard palate	5 (10.0)
STAGING	
TISN0M0 (stage 0)	25 (50.0)
T1N0M0 (stage 1)	1 (2.0)
T2N0M0 (stage 2)	0 (0.0)
T3N0M0 (stage 3)	1 (2.0)
T3N1M0 (stage 3)	3 (6.0)
T4N0M0 (stage 4)	1 (2.0)
T4N1M0 (stage 4)	19 (38.0)
GRADING	
G1	13 (26.0)
G2	31 (62.0)
G3	5 (10.0)
G4	1 (2.0)
ORAL POTENTIALLY MALIGNANT DISORDERS	29 (58.0)
SURGICAL TREATMENT OF PRIMARY OSCC	
Local tumor resection	33 (66.0)
Hemiglossectomy	10 (20.0)
Maxillary Osteotomy	4 (8.0)
Hemimandibulectomy	3 (6.0)
Cervical neck dissection	21 (42.0)
CHEMOTHERAPY	1 (2.0)
RADIOTHERAPY	8 (16.0)

N° OF PATIENTS WITH LOCAL RECURRENCES	30 (60.0)
N° OF SECONDARY SURGICAL LOCAL RESECTIONS	Mean ± SD (Range)
	1.74 ± 2.18 (1−9)
ECOG	
Status 0	33 (66.0)
Status 1	17 (34.0)

**Table 4 cancers-13-01855-t004:** Differences in sleep quality, anxiety, depression and pain in 50 OSCC patients and 50 controls.

	OC Survivors	Controls	*p*-Value
PSQI Cronbach Alpha	0.76	0.91	
	Median-IQR	Median-IQR	
PSQI			
Subjective sleep quality	6; [3–9]	4; [3–5]	0.017 *
Sleep latency	1; [1–2]	1; [0–1]	<0.001 **
Sleep duration	1; [0–2]	0; [0–1]	0.029 *
Habitual sleep efficiency	1; [0–2]	1; [0–1]	0.512
Sleep disturbances	0; [0–2]	0; [0–1]	0.400
Use of sleep medications	1; [1–2]	1; [1–1]	0.740
Daytime dysfunction	0; [0–1]	0; [0–0]	0.004 **
HAM-A	12; [9–24]	5; [3–6]	<0.001 **
HAM-D	10; [6–24]	4; [3–6]	<0.001 **
ESS	5; [2–9]	3; [3–4]	0.001 **
NRS	2; [0–5]	0; [0–0]	<0.001 **
T-PRI	2; [0–9]	0; [0–0]	<0.001 **

Legend: ESS = Epworth Sleepiness Scale; HAM-A = Hamilton Anxiety Scale; HAM-D = Hamilton Depression Scale; IQR = interquartile range. NRS = Numeric Rating Scale; McGill: PSQI = Pittsburgh Sleep Quality Index; T-PRI: Total Pain Rating Index. The significance difference between medians was measured by the Mann–Whitney test. * Significant 0.01 ≤ *p* ≤ 0.05 ** Significant *p* ≤ 0.01.

**Table 5 cancers-13-01855-t005:** Correlation analysis between the PSQI items and anxiety, depression and pain in 50 OSCC patients and 50 controls.

	HAM-A		HAM-D		ESS		NRS		T-PRI	
	Rho	*p-*Value	Rho	*p-*Value	Rho	*p-*Value	Rho	*p-*Value	Rho	*p-*Value
PSQI	,671	<0.001 **	,735	<0.001 **	,242	0.138	,250	0.125	,374	0.019 *
Subjective sleep quality	,423	0.007 **	,528	0.001 **	,078	0.636	-,023	0.891	,251	0.124
Sleep latency	,305	0.059	,470	0.003 **	,285	0.079	,172	0.295	,181	0.271
Sleep duration	,488	0.002 **	,572	<0.001 **	,139	0.398	,206	0.209	,216	0.187
Habitual sleep efficiency	,542	<0.001 **	,573	<0.001 **	-,004	0.981	,194	0.237	,232	0.155
Sleep disturbances	,480	0.002 **	,599	<0.001 **	,102	0.535	,189	0.249	,395	0.013 *
Use of sleep medications	,298	0.066	,149	0.364	,003	0.984	,167	0.309	,051	0.760
Daytime dysfunction	,561	<0.001 **	,506	0.001 **	,461	0.003 **	,118	0.473	,389	0.014 *

Legend: ESS = Epworth Sleepiness Scale; HAM-A = Hamilton Anxiety Scale; HAM-D = Hamilton Depression Scale; NRS = Numeric Rating Scale; McGill: PSQI = Pittsburgh Sleep Quality Index; T-PRI: Total Pain Rating Index. Correlation between PSQI items and other variables was measured with the Spearman correlation analysis. * Moderately significant 0.01 < *p* ≤ 0.05; ** strongly significant *p* ≤ 0.01.

**Table 6 cancers-13-01855-t006:** Multiple linear regression analysis predicting poor QoS (PSQI > 5) in 50 OC survivors.

	Model 1		Model 2		Model 3		Model 4		Model 5		Model 6	
Parameter	Beta	*p*- Value	Beta	*p*- Value	Beta	*p*- Value	Beta	*p*- Value	Beta	*p*- Value	Beta	*p*- Value
	(SE)		(SE)		(SE)		(SE)		(SE)		(SE)	
Gender	−2.93	0.094	−4.12	0.029	−1.44	0.354	−2.85	0.113	−2.46	0.168	−2.55	0.226
Male vs. Female	(1.70)		(1.78)		(1.53)		(1.74)		(1.74)		(1.04)	
Age	0.04	0.583	−0.03	0.739	0.02	0.618	0.04	0.546	0.04	0.522	0.07	0.399
	(0.06)		(0.08)		(0.05)		(0.07)		(0.06)		(0.07)	
Education	−0.29	0.042 *	−0.24	0.207	−0.11	0.364	−0.30	0.045 *	−0.21	0.155	−0.09	0.579
	(0.14)		(0.18)		(0.12)		(0.14)		(0.14)		(0.17)	
Employment	1.41	0.511	1.94	0.376	0.53	0.788	1.61	0.477	0.97	0.659	2.57	0.277
Yes vs. No	(2.12)		(2.15)		(1.79)		(2.25)		(2.17)		(2.30)	
Married	−2.51	0.110	−1.42	0.425	−1.30	0.327	−2.70	0.114	−2.83	0.092	−2.58	0.186
Yes vs. No	(1.52)		(1.75)		(1.31)		(1.66)		(1.62)		(1.88)	
BMI	0.10	0.442	−0.02	0.871	0.11	0.311	0.11	0.419	0.12	0.429	−0.09	0.634
	(0.13)		(0.16)		(0.11)		(0.13)		(0.14)		(0.19)	
Smoking	0.71	0.715	3.89	0.098	1.25	0.455	0.60	0.765	0.94	0.631	2.82	0.184
Yes vs. No	(1.94)		(1.52)		(1.65)		(1.99)		(1.94)		(2.05)	
Alcohol consumption	2.54	0.079	3.90	0.018 *	1.95	0.135	2.54	0.084	2.63	0.071	3.32	0.077
Yes vs. NO	(1.40)		(1.52)		(1.27)		(1.42)		(1.40)		(1.78)	
Potentially malignant disorders			2.66	0.097							1.63	0.330
Yes vs. No			(1.54)								(1.63)	
Number of operations			0.16	0.613							0.20	0.497
			(0.32)								(0.29)	
Radiotherapy			1.52	0.534							−0.56	0.852
Yes vs. NO			(2.40)								(3.01)	
T3N0M0 vs. TISN0M0			1.71	0.486							2.61	0.327
			(2.41)								(2.59)	
T4N0M0 vs. TISN0M0			1.13	0.497							0.67	0.773
			(1.64)								(2.29)	
Medications			−10.2	0.045 *							−7.32	0.193
Yes vs. NO			(4.98)								(5.42)	
HAM-A					0.08	0.352					0.09	0.499
					(0.08)						(0.13)	
HAM-D					0.15	0.001 **					0.16	0.001 **
					(0.06)						(0.07)	
ESS							−0.04	0.749			−0.13	0.378
							(0.14)				(0.15)	
NRS									0.06	0.853	0.30	0.374
									(0.33)		(0.33)	
T-PRI									0.09	0.423	−0.11	0.431
									(0.11)		(0.13)	
R2 Adjusted	31.7%		37.9%		52.1%		29.6%		31.7%		44.3%	
ΔR2 Adjusted	31.7%		6.2%		20.4%		−2.1%		0.0%		12.6%	
	(*p* = 0.009 **)		(*p* = 0.222)		(*p* < 0.001 **)		(*p* = 0.749)		(*p* = 0.377)		(*p* = 0.043 *)	

SE are the standard errors of the beta estimates. The *p*-values were obtained from the hypothesis test on the regression coefficients. * Moderately significant 0.01 < *p*-value ≤ 0.05 ** Strongly significant *p*-value ≤ 0.01. Legend: ESS = Epworth Sleepiness Scale; HAM-A = Hamilton Anxiety Scale; HAM-D = Hamilton Depression Scale; NRS = Numeric Rating Scale; McGill: PSQI = Pittsburgh Sleep Quality Index; T-PRI: Total Pain Rating Index.

## Data Availability

The data presented in this study are available on request from the corresponding author. The data are not publicly available due to patient sensitive data.

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
