# Peer review of "Sleep Disorders and Psychological Profile in Oral Cancer Survivors: A Case-Control Clinical Study"

_cancers, 2021, doi:10.3390/cancers13081855_

Round 1

Reviewer 1 Report

  1. Please elaborate on reasons for exclusion criteria ( patients with HPV, patients who experience severe side effects, specifically hyposalivation. This may introduce bias in your outcome. In other words, if you excluded patients with more severe symptoms you will have patients having less severe sleep disorders. For example, your findings were different from Shuman et al ref:25, mostly because you excluded patients with hypo-salivation to start with. 
  2. Your study is underpowered, which is definitely affecting your outcome, specially when compare to other authors. In statistics term, you usually need 10 patients per each variable for your data to have a significant value, you have about 16-17 variables.. Address impact of low powered study in your outcome. 
  3. Provide reference for sentence on page 11, line 386 “ women are more vulnerable to the stress related to a cancer diagnosis .......”
  4. Table 4. Precancerous lesions 29%. Where are they in the text, what was the diagnosis, no clear.

  5. page 12, line 440. I don’t particularly agree with your statement “ all patients were recruited at a single...” in fact the information gathered by your study may very well represent the population of this particular region in Naples. Line 441 you introduced the concept of a cross sectional study ? Where as in the title and under material and methods you described it as a case-control study??? In fact, you do a case-control study to prove temporality and causality! Please explain 

Author Response

Reviewer 1

1) Please elaborate on reasons for exclusion criteria (patients with HPV, patients who experience severe side effects, specifically hyposalivation. This may introduce bias in your outcome. In other words, if you excluded patients with more severe symptoms you will have patients having less severe sleep disorders. For example, your findings were different from Shuman et al ref:25, mostly because you excluded patients with hypo-salivation to start with. 

Thank you for your valuable comment.  We agree with this reviewer’s opinion on the potential underestimation of sleep disorders in this category of patients. However, our aim was to estimate the prevalence of sleep disorders in OC survivors regardless of the severe and permanent side effects induced by radiotherapy. Indeed, as you pointed out, radiotherapy side effects such as hypo-salivation, fibrosis, mouth restriction etc. may be per se causes of sleep disorders due to their impact on everyday life activities. On the contrary, it may be not obvious the presence of sleep impairment in a case of OC survivor without such complications. Similarly, HPV-related OSCC is generally treated by means of chemotherapy/radiotherapy, while the OSCC non-HPV-related needs primarily surgical treatment which often hesitate in local post-surgical disablement depending on the extension of the tumor resection. Moreover, conversely to HPV-OSCC patients, most OC survivors, are affected by oral potentially malignant disorders, especially those with field cancerization, which obliged to several follow-up visits and repetitive surgical biopsies over the follow-up period in order to exclude the presence of a relapse or, worse, a secondary tumor. In other words, we selected a sample as much homogenous as possible in order to not introduce any potential obvious confounders. Indeed, in our sample, sleep disorders were found to be statistically significantly more prevalent in the case group compared with the control group, and regression analyses could explain the 44.3% of the variance, which means that there should be other factors (probably, biological, psychological, etc) which may impair quality of sleep. We recognize the potential underestimation due to the inclusion criteria, therefore we have acknowledged it in the limitation section, Page 15, paragraph 3.

2) Your study is underpowered, which is definitely affecting your outcome, specially when compare to other authors. In statistics term, you usually need 10 patients per each variable for your data to have a significant value, you have about 16-17 variables.. Address impact of low powered study in your outcome. 

Thank you for your comment. We have recognized this point as a limitation of the study in the discussion section. Page 15.

3)Provide reference for sentence on page 11, line 386 “ women are more vulnerable to the stress related to a cancer diagnosis .......”.

Thank you for your comment. We have added the appropriate reference.

4) Table 4. Precancerous lesions 29%. Where are they in the text, what was the diagnosis, no clear.
Thank you for this comment and we apologize for the poor clarity. We have better specified this point in the text. Page 9, Lines 28-29.

5) page 12, line 440. I don’t particularly agree with your statement “all patients were recruited at a single...” in fact the information gathered by your study may very well represent the population of this particular region in Naples. Line 441 you introduced the concept of a cross sectional study? Where as in the title and under material and methods you described it as a case-control study??? In fact, you do a case-control study to prove temporality and causality! Please explain 

 Thank you for your comment.  According to your suggestion, we have deleted the first sentence. With respect to the second comment, we apologize for the mistake. As per design of this case-control study, the measurements of sleep disorders, anxiety, depression and pain were taken just once and were not repeated over the time. Therefore, we could explore the differences in terms of sleep disorders, anxiety and depression prevalence between the two groups; however, we could not explore the temporality and causality of these variables. Page 15, line 8.

Reviewer 2 Report

Psychological disorders have sometimes been overlooked in head and neck cancer patients but they require a proper and attentive management, and reports dealing with this topic are thus welcome. The authors of the present case-control clinical study focus on sleep disorders. The manuscript is well-prepared containing all necessary parts, it is written in very good English, and refers to previously published data. One of the strengths is that the authors provide detailed information on comorbidities and medication. However, I suspect a methodological error which I address below.

Major comment:

1) The two groups were matched only for age and sex. However, there are significant imbalances between the two groups in terms of comorbidities and medication. Both of these aspects impact on psychological well-being and quality of sleep. Therefore, the sleep disorders found in oral cancer survivors might not be due to the oncologic diagnosis but due to the many other diseases the patients had.

Minor comments:

1) In the introduction when talking about cancer patients (including cited material), please mention whether these are also survivors or patients in the course of their treatment.

2) You mention „independent of the cancer characteristics, cancer site, staging and grading of the malignancy, type of treatment (surgery, chemotherapy or radiotherapy), pain and presence of potentially malignant disorders.“ Did you include all these characteristics in the multivariate model? If yes, why do they not appear in Table 6?

2) Please reconsider the use of the word „contrast“ in the following situations:

- line 366: you did no detect xerostomia, thus it is difficult to compare it with a study which probably included differently treated patients and analysed this side effect

- line 377, the same issue, what you write makes an impression that insomnia and hypersomnolence might not be related to chemotherapy and radiotherapy, however, it cannot be stated in such a way because you did not properly investigate this correlation

- line 392: can you really say that based on 8 patients „BMI cannot be considered a predictor of poor sleep“?

3) Please consider rephrasing „but could also be considered in some cases a variable to evaluate independently“ to better explain what you mean.

4) There are already a lot of abbreviations, to improve readability, do not use them for sleep disorders because SD in oncology is very often used for „stable disease“ and instead of using OCS consider “OC survivors“.

5) Please do not use the letter O instead of zero 0, e.g. in Table 6 when mentioning the TNM classification.

Author Response

Psychological disorders have sometimes been overlooked in head and neck cancer patients but they require a proper and attentive management, and reports dealing with this topic are thus welcome. The authors of the present case-control clinical study focus on sleep disorders. The manuscript is well-prepared containing all necessary parts, it is written in very good English, and refers to previously published data. One of the strengths is that the authors provide detailed information on comorbidities and medication. However, I suspect a methodological error which I address below.

Major comment:

1) The two groups were matched only for age and sex. However, there are significant imbalances between the two groups in terms of comorbidities and medication. Both of these aspects impact on psychological well-being and quality of sleep. Therefore, the sleep disorders found in oral cancer survivors might not be due to the oncologic diagnosis but due to the many other diseases the patients had.

Thank you for your valuable comment. The prevalence of systemic comorbidities was overall quite balanced between the two groups with the exception of hypercholesterolemia, hypertension, and gastric disease (we have reviewed medical charts and all the patients with gastro-intestinal diseases suffered from gastritis). Consequently, the related medications were different as well, and were slightly correlated with sleep disorders in the model 2 (p-value=0.045) of the regression analyses. However, this parameter did not result in a strongly significant increase in the R2 (∆R2 = 6.2%, p-value:<0.222). Therefore, we found that medications did not add significance to sleep disorders based on the final regression analysis, and we think medications may not have a pivotal role in explaining the higher prevalence of sleep disorders in in this group of OC survivors. According to your interesting comment we have addressed this consideration in the text, page 11 lines 4-5.

Minor comments:

2) In the introduction when talking about cancer patients (including cited material), please mention whether these are also survivors or patients in the course of their treatment.

Thank you for your comment. We have referred to studies on cancer survivors. In the case a study mentioned both patients, those under treatment and those who survived, we just took the data on the OC survivors.

3) You mention „independent of the cancer characteristics, cancer site, staging and grading of the malignancy, type of treatment (surgery, chemotherapy or radiotherapy), pain and presence of potentially malignant disorders.“ Did you include all these characteristics in the multivariate model? If yes, why do they not appear in Table 6?.

Thank you for your comment. We apologize for the mistake, we did not include cancer site, grading and chemotherapy (this last one since only one patient was treated with chemotherapy). All the others are displayed in table 6 (staging, surgery, radiotherapy, pain which was assessed by NRS and T-PRI, potentially malignant disorders). We have made appropriate changes in the text.

4) Please reconsider the use of the word „contrast“ in the following situations:

- line 366: you did no detect xerostomia, thus it is difficult to compare it with a study which probably included differently treated patients and analysed this side effect.

Thank you for your comment. We have accordingly changed this sentence as per your suggestion; page 13, lines 2-6.

- line 377, the same issue, what you write makes an impression that insomnia and hypersomnolence might not be related to chemotherapy and radiotherapy, however, it cannot be stated in such a way because you did not properly investigate this correlation.

Thank you for your comment. We agree on this point and have better rephrased this sentence, page 13, lines 18-20.

- line 392: can you really say that based on 8 patients „BMI cannot be considered a predictor of poor sleep“?

Thank you for this comment. Notoriously, an altered BMI could predict poor quality of sleep. However, in our sample, none of these 8 patients suffered from OSAS. Moreover, according to the univariate and multivariate analyses, no significant differences were detected between the two groups in terms of BMI. According to your valuable suggestion we have rephrased this sentence, page 13, lines 1-2.

5) Please consider rephrasing but could also be considered in some cases a variable to evaluate independently“ to better explain what you mean.

Thank you for your comment. We have rephrased the sentence. Page 14 lines 22-23.

6) There are already a lot of abbreviations, to improve readability, do not use them for sleep disorders because SD in oncology is very often used for „stable disease“ and instead of using OCS consider “OC survivors“.

Thank you for your comment. We have accordingly changed as per your suggestions.

7) Please do not use the letter O instead of zero 0, e.g. in Table 6 when mentioning the TNM classification.

Thank you for your comment, we have checked table 6.

Reviewer 3 Report

  • This article is very relavant and well-written. There is only five changes to consider. 
  • What do you mean by "psychological profile"? If you evaluate anxiety, depression and quality of life, pain and sleepiness, it is better to talk about "psychopathological profile" or "psychiatric profile". The title should be change, just as mentioned in the keywords. The same thing happens in the discussion where authors mention "psychological symptoms". 
  • Simple summary, abstract and introduction are well written and really relevant.
  • In the procedure paragraph, there is mention "social habits" but it only refers to smoking and alcohol consumption such as in Table 1 and in the first results sentence. Authors should be more specific.
  • In Table 6, the ΔR2 Adjusted and the R2 Adjusted are the most important results so they should be mentioned in the first lines. 
  • In the results, authors states that the correlation with alcohol consumption is p-value: 0.019* but it does not appear in the Table 6.
  • The Statistical analysis is a little light. Is it possible to explain what is important in the R2 adjusted? And to explain how you work with the qualitative data (tabac and alcohol) in the multiple regression analysis?

Author Response

Reviewer 3

This article is very relavant and well-written. There is only five changes to consider. 

  • What do you mean by "psychological profile"? If you evaluate anxiety, depression and quality of life, pain and sleepiness, it is better to talk about "psychopathological profile" or "psychiatric profile". The title should be change, just as mentioned in the keywords. The same thing happens in the discussion where authors mention "psychological symptoms". 

Thank you for your comment. This study has evaluated some psychological aspects worth of consideration in OC survivors and, therefore, we expect it could be of interest also for psychiatrists. According to your thoughtful suggestion, we have added this term “psychiatric profile” in the key words in order to reach also the category of psychiatrists, however we would feel more comfortable to use the terminology “psychological profile” in the title and in the text because psychiatric disorders include other diseases that we did not evaluate in this study.

  • Simple summary, abstract and introduction are well written and really relevant.
  • In the procedure paragraph, there is mention "social habits" but it only refers to smoking and alcohol consumption such as in Table 1 and in the first results sentence. Authors should be more specific.

Thank you for your comment. We have accordingly modified “social habits in “risk factors”.

  • In Table 6, the ΔR2 Adjusted and the R2 Adjusted are the most important results so they should be mentioned in the first lines. 

Thank you for your comment. We would rather keep the present design of the table.

  • In the results, authors states that the correlation with alcohol consumption is p-value: 0.019* but it does not appear in the Table 6.

Thank you for your comment. We apologize for the mistake. We have accordingly changed it in the results section, page10, line 33

  • The Statistical analysis is a little light. Is it possible to explain what is important in the R2 adjusted? And to explain how you work with the qualitative data (tabac and alcohol) in the multiple regression analysis?

Thank you for your comment. The Author A.M, the statistician, has provided detailed explanation on this point at page 8, line 7-10

Round 2

Reviewer 2 Report

I thank the authors for addressing my comments. However, I'm not satisfied with their response to the major comment for the following reasons:

1) In the revised version, there are no changes on page 11, lines 4-5. I found the following "...was positively correlated with alcohol consumption (p-value: 0.018*) and poorly with the use of systemic medications (p-value: 0.045*) but did not result in a significant increase in the R2 value (∆R2 = 6.2%, p-value:0.222 in the model 2.:..." on lines 291-293, so maybe they are referring to this. In your further revisions, please state clearly where the revision can be found in the text.

2) Stating that a p-value of 0.045 poorly correlates, makes an impression as if the authors wanted to somehow steer the results. Based on what the authors say the correlation is poor? Either it is statistically significant or not. And if it is so, then it should be put into context accompanied by a sufficient interpretation.

3) The reasoning the authors put in their reply, should also be given in the text. The readers may ask a similar question.

4) The authors should also explain why in one model comorbidities have a significant value and in the other one it is not the case. Why did the authors finally choose the latter model and no the former one?

5) The authors should also expand the discussion to cover possible relationships of comorbities and medication with sleep disturbances, particularly those not balanced in their report.

Author Response

REVIEWER 2

I thank the authors for addressing my comments. However, I'm not satisfied with their response to the major comment for the following reasons:

  • In the revised version, there are no changes on page 11, lines 4-5. I found the following "...was positively correlated with alcohol consumption (p-value: 0.018*) and poorly with the use of systemic medications (p-value: 0.045*) but did not result in a significant increase in the R2 value (∆R2 = 6.2%, p-value:0.222 in the model 2.:..." on lines 291-293, so maybe they are referring to this. In your further revisions, please state clearly where the revision can be found in the text.
  • Stating that a p-value of 0.045 poorly correlates, makes an impression as if the authors wanted to somehow steer the results. Based on what the authors say the correlation is poor? Either it is statistically significant or not. And if it is so, then it should be put into context accompanied by a sufficient interpretation.

Answer to comments 1 and 2: Thank you for your comment. We apologize for the lack of clarity in referring to the changes made and also for the inappropriate use of “poorly”. We have better explained the results of the analysis (Page 11, lines 1-6). Moreover, we have further specified that in the final model (model 6) depression was the most contributive factor (page 11, lines 12 and 15-16).  This final consideration was also added in the discussion section (page 14, lines 25-26) and conclusion (page 16, line 3). Please, find our new changes in red, while the previous are highlighted in yellow.

  • The reasoning the authors put in their reply, should also be given in the text. The readers may ask a similar question.

Thank you for your comment; following your suggestion with regard to comorbidities and medications, we have given explanations also in the discussion section (page 13, lines 10-14).

  • The authors should also explain why in one model comorbidities have a significant value and in the other one it is not the case. Why did the authors finally choose the latter model and no the former one?

Thank you for your comment. We reported results from both the models. Indeed, as reported in the previous answer to comments 1-2, we have explained both model 2 and model 6 (which is the final model of the regression analyses). Overall, although in the second model we found a positive correlation between medications (in addition to alcohol) and sleep disorders increasing the ∆R2 of 6.2% of the variance of the PSQI, in the final full model, where all the variables where entered simultaneously, medications did not show any statistically significant correlation while only depression showed a p-value <0.001, strongly contributing to poor sleep.

  • 5) The authors should also expand the discussion to cover possible relationships of comorbities and medication with sleep disturbances, particularly those not balanced in their report.

Thank you for your comment. We have expanded this point in the discussion section (page 13, lines 2-7 and 10-14)